Corrected: Publisher correction

# Imaging the square of the correlated two-electron wave function of a hydrogen molecule

M. Waitz[1], R.Y. Bello[2], D. Metz[1], J. Lower[1], F. Trinter[1], C. Schober[1], M. Keiling[1], U. Lenz[1], M. Pitzer [3],
K. Mertens[4], M. Martins[4], J. Viefhaus[5], S. Klumpp[6], T. Weber[7], L.Ph.H. Schmidt[1], J.B. Williams[8], M.S. Schöffler[1],
V.V. Serov[9], A.S. Kheifets[10], L. Argenti[2,13], A. Palacios[2], F. Martín[2,11,12], T. Jahnke[1] & R. Dörner [1]

The toolbox for imaging molecules is well-equipped today. Some techniques visualize the geometrical structure, others the electron density or electron orbitals. Molecules are many-body systems for which the correlation between the constituents is decisive and the spatial and the momentum distribution of one electron depends on those of the other electrons and the nuclei. Such correlations have escaped direct observation by imaging techniques so far. Here, we implement an imaging scheme which visualizes correlations between electrons by coincident detection of the reaction fragments after high energy photofragmentation. With this technique, we examine the $H_2$ two-electron wave function in which electron–electron correlation beyond the mean-field level is prominent. We visualize the dependence of the wave function on the internuclear distance. High energy photoelectrons are shown to be a powerful tool for molecular imaging. Our study paves the way for future time resolved correlation imaging at FELs and laser based X-ray sources.

[1] Institut für Kernphysik, J. W. Goethe Universität, Max-von-Laue-Str. 1, 60438 Frankfurt, Germany. [2] Departamento de Química, Universidad Autónoma de Madrid, 28049 Madrid, Spain. [3] Universität Kassel, Heinr.-Plett-Strasse 40, 34132 Kassel, Germany. [4] Institut für Experimentalphysik, Universität Hamburg, Luruper Chaussee 149, 22761 Hamburg, Germany. [5] FS-PE, Deutsches Elektronen-Synchrotron DESY, Notkestrasse 85, 22607 Hamburg, Germany. [6] FS-FLASH-D, Deutsches Elektronen-Synchrotron DESY, Notkestrasse 85, 22607 Hamburg, Germany. [7] Chemical Sciences Division, Lawrence Berkeley National Laboratory, Berkeley, CA 94720, USA. [8] Department of Physics, University of Nevada Reno, 1664 N. Virginia Street, Reno, NV 89557, USA. [9] Department of Theoretical Physics, Saratov State University, 83 Astrakhanskaya, Saratov, 410012, Russia. [10] Research School of Physical Sciences, The Australian National University, Canberra, ACT 0200, Australia. [11] Instituto Madrileo de Estudios Avanzados en Nanociencia, 28049 Madrid, Spain. [12] Condensed Matter Physics Center (IFIMAC), Universidad Autónoma de Madrid, 28049 Madrid, Spain. [13] Present address: Department of Physics and CREOL College of Optics & Photonics, University of Central Florida, Orlando, FL 32816, USA. Correspondence and requests for materials should be addressed to F.M. (email: fernando.martin@uam.es) or to R.D. (email: doerner@atom.uni-frankfurt.de)

maging the wave function of electrons yields detailed information on the properties of matter, accordingly, experiments have pursued this goal since decades. For example, in solid state physics photoionization is routinely used as a powerful tool[1] for single-electron density imaging. Photon-based techniques have the particular strength that they can be in principle implemented in pump-probe experiments opening additionally the perspective to go from still images to movies. For atoms and molecules photoionization has also been proposed as a promising technique to image orbitals[2], but no positive outcomes were reported so far. The reverse process of photoionization, namely high harmonic generation, has succeeded in accomplishing this goal of orbital imaging[3]. Further techniques for imaging molecular orbitals are electron momentum spectroscopy[4] or strong field tunnel ionization[5].

While the toolbox to image single electrons is well equipped, endeavors to directly examine an entangled two-electron wave function have, so far, not been successful and corresponding techniques are lacking. This is particularly unfortunate, as electron correlation which shapes two-electron wavefunctions is of major importance across physics and chemistry. It is electron correlation which is at the heart of fascinating quantum effects such as superconductivity[6] or giant magnetoresistance[7]. Even in single atoms or molecules, electron correlation plays a vital role and continues to challenge theory. For example, the single-photon double ionization, i.e., the simultaneous emission of two electrons after photoabsorption, is only possible due to electron–electron correlation effects, as the photon cannot interact with two electrons simultaneously. Instead, the second electron is emitted either after an interaction with the first electron (which is typically described as a "knock-off process") or because of the initial entanglement of the two electrons due to electron correlation prior to the absorption of the photon (in a process termed "shake off")[8]. Although the importance of electron correlation is intuitively understandable in processes which obviously involve two electrons, it turns out, that even for bound stationary states of atoms and molecules, electron correlation contributions are crucial: within the commonly used Hartree–Fock approximation, the calculated values of binding energies are often in no satisfying accordance to those actually occurring in nature. Here the basic cause is that the Hartree–Fock approximation is a mean-field theory, which considers only an overall mean potential generated by the ensemble of electrons, and as such neglects electron–electron correlation by definition.

In this manuscript, we show that the correlated molecular wave function can be visualized by the simultaneous use of two well-established and well-understood methods: photoelectron emission on the one hand and coincident detection of reaction fragments on the other hand. Our novel experimental approach allows us to visualize the square of the $H_2$ correlated two-electron wave function. In the ionization step, one of the electrons is mapped onto a detector and simultaneously the quantum state of the second electron is determined by coincident detection of the fragments.

## Results

**Concept of correlation imaging.** The properties of a photo-ionization event, given by the ionization amplitude $D$, are determined (within the commonly used dipole approximation) by only three ingredients: the initial state of the system $\phi_0$, which we want to image, the properties of the dipole operator $\hat{\mu}$ (responsible for the photoionization) and the final state representing the remaining cation and a photoelectron with momentum $\mathbf{k}$, $\chi_\mathbf{k}$:

$$D = \int \phi_0(\mathbf{r})\hat{\mu}(\mathbf{r})\chi_\mathbf{k}(\mathbf{r})d\mathbf{r}, \qquad (1)$$

where $\mathbf{r}$ represents the coordinates of target electrons. The initial wave function is directly accessible provided that the other two constituents do not introduce significant distortions. This is the case when utilizing circularly polarized light and examining high energy electrons (Born limit) within the polarization plane. As an illustration, let us consider the one-electron $H_2^+$ molecular ion. At a high enough energy, the continuum electron can be described by a plane wave. In this case, the photoionization differential cross section in the electron emission direction $(\theta, \varphi)$ (the so-called molecular frame photoelectron angular distribution, MFPAD) is simply proportional to the square of the Fourier transform (FT) of the initial state, $\phi_0(\mathbf{k})$ (see methods section):

$$\frac{dP}{d(\cos\theta)dk} = k^2(2\pi)^{3/2}\left|\frac{1}{2\pi^{3/2}}\int\phi_0(\mathbf{r})e^{i\mathbf{k}\mathbf{r}}d\mathbf{r}\right|^2 \qquad (2)$$

$$= k^2(2\pi)^{3/2}|\phi_0(\mathbf{k})|^2. \qquad (3)$$

Here $\theta$ denotes the polar angle with respect to the molecular axis and $\varphi$ the corresponding azimuthal angle. Thus, by choosing high-photon energies and restricting the measurement of the MFPAD to the polarization plane ($\varphi = 90°$ and $270°$) of the circularly polarized light, the initial electronic wave function is directly mapped onto the emitted photoelectron. Figure 1 illustrates this mapping procedure for the ground state of $H_2^+$ (Fig. 1a: electronic wave function in coordinate space; Fig. 1b the square of the Fourier transform of Fig. 1a; Fig. 1c the same in logarithmic color scale). As can be seen from Fig. 1d, the MFPAD for an

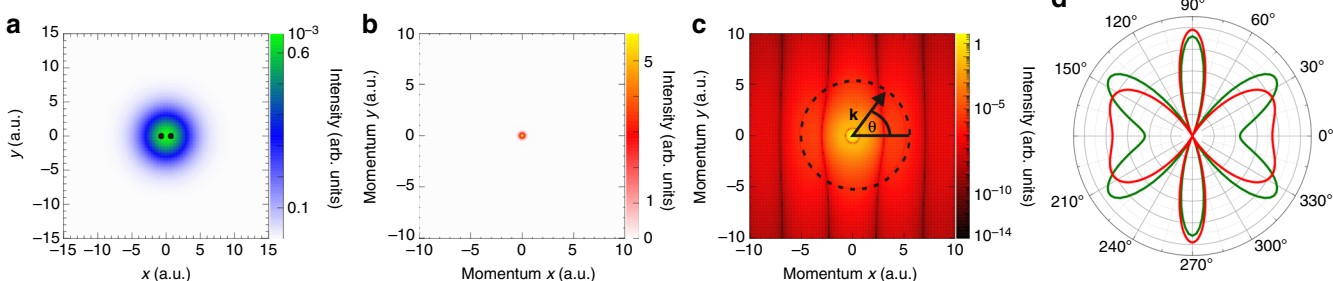

**Fig. 1** Imaging of the $H_2^+$ one-electron wave function. **a** The electronic wave function of $H_2^+$ in the polarization plane for an internuclear distance $R = 1.4$ a.u. The positions of the two nuclei are indicated by black dots. **b** The square of the Fourier transform of a in the $(k_x, k_y)$ plane. **c** The same as (**b**), but in logarithmic color scale. Notice the appearance of nearly vertical fringes, when $|\mathbf{k}|$ is significantly different from zero. The approximate periodicity of these fringes is $\Delta k_x \sim 2\pi/R$. The dashed line indicates the region of momentum space associated with an electron kinetic energy of 380 eV (i.e., a radius of $|\mathbf{k}| = 5.3$ a.u.) and $\theta$ is the angle with respect to the molecular axis. **d** Polar plot of the intensity distribution in **c** along the dashed line (red) and the corresponding MFPAD in the plane of polarization of the ionizing radiation obtained from nearly exact calculations (green)

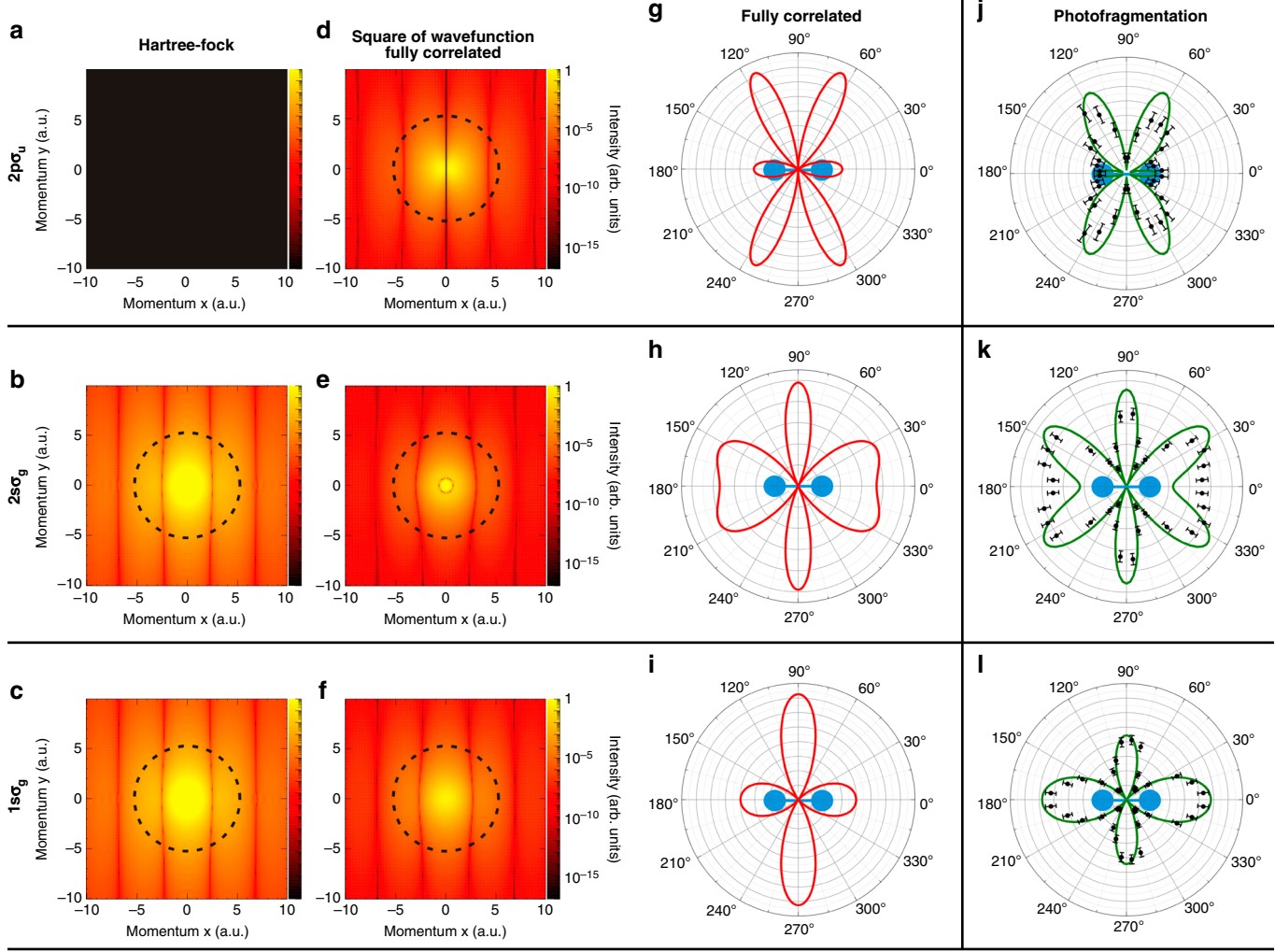

**Fig. 2** Correlation imaging of the $H_2$ two-electron wave function. **a–f** Momentum distributions of electron A resulting from the projection of the two-electron wave function of $H_2$ onto different $H_2^+$ states of electron B; **a**, **c** uncorrelated Hartree-Fock wave function; **d**, **f** fully correlated wave function. The different quantum states of electron B are $2p\sigma_u$ (**a**, **d**), $2s\sigma_g$ (**b**, **e**) and $1s\sigma_g$ (**c**, **f**). Circular lines show $|\mathbf{k}| = 5.3$ a.u. (**c**, **d**, **f**) and $|\mathbf{k}| = 5.2$ a.u. **b**, **e** which correspond to ionization by a photon of 400 eV energy. **g–i** ground state wave function (intensity distributions along the circular lines shown in (**d**, **f**). **j–l** Experimental and theoretical MFPADs (symbols and green line, respectively) obtained after photoionization with circularly polarized photons of an energy of 400 eV for the same final states of electron B measured in coincidence. Ions and electrons are selected to be in the plane of polarization of the ionizing photon and data for left and right circularly polarized light are added. Molecular orientation as indicated. The error bars indicate the standard deviation of the mean value

electron of 380 eV is very similar to $|\phi_0(\mathbf{k})|^2$ for the chosen momentum $\mathbf{k}$ (the square of the FT along the dashed line shown in Fig. 1c). Notice that, due to the smallness of the cross section at such high electron momentum, the main features of the FT are only apparent in the logarithmic plot shown in Fig. 1c.

This tool of high energy photoelectron imaging can now be combined with coincident detection of the quantum state of a second electron to visualize electron correlation in momentum space. We dissect the entangled two-electron wave function by analyzing a set of conditional angularly resolved cross sections corresponding to a high energy continuum electron (A) and a bound electron (B) detected in a different region of the two-electron phase space. Quantum mechanically, this is equivalent to projecting the initial two-electron wave function onto products of different $H_2^+$ (bound) molecular orbitals (B) and a plane wave (A) (see Methods section). In doing so, one can thus determine if and how the density distribution of one electron changes upon changing the region of phase space in which one detects the other, correlated, electron.

**Application on $H_2$.** Figure 2 illustrates this concept and highlights the differences between the uncorrelated Hartree–Fock wave function and the highly correlated nearly exact wave function. The corresponding one-electron momentum distributions resulting from the projection of the corresponding ground state wave functions onto different states of the bound electron B, $n_\lambda$, are depicted in Fig. 2a–c (Hartree–Fock) and Fig. 2d–f (exact) as functions of the momentum components parallel ($k_{x,A}$) and perpendicular ($k_{y,A}$) to the molecular axis. The different rows correspond to the different states in which the second electron B is left after photoionization, i.e., they correspond, from bottom to top, to projections of the ground state wave function onto the $n_\lambda = 1s\sigma_g$, $2s\sigma_g$, and $2p\sigma_u$ states of $H_2^+$. Thus, as in our one-electron example shown in Fig. 1, the different panels in Fig. 1 contain direct images of different pieces of the ground state of $H_2$ through the square of the corresponding FTs. The role of electron correlation is quite apparent in this presentation: Fig. 1a is empty for the uncorrelated Hartree–Fock wave function, since projection of the latter wave function onto the $2p\sigma_u$ orbital is exactly

zero, while this is not the case for the fully correlated wave function (Fig. 1d); also, Fig. 1b, c for the uncorrelated description are identical, while Fig. 2e and f for the correlated case are significantly different. As in the example of Fig. 1c, a fixed energy corresponds to points around the circumference of a circle. The density distributions pertaining to points around the circles of Fig. 2a–f are shown in Fig. 2g–i.

Experimentally, these conditional probabilities are obtained by measuring in coincidence the momentum of the ejected electron and the proton resulting from the dissociative ionization reaction

$$\gamma(400\,\text{eV}) + H_2 \rightarrow H_2^+(n_\lambda) + e^- \qquad (4)$$

$$\searrow$$

$$H(n) + H^+, \qquad (5)$$

which, as explained below, allows us to determine the final ionic state characterized by the quantum number $n_\lambda$. Fig. 2j–l depicts the experimental results of the measured angular distributions of electron A together with numerical data resulting from a nearly exact theoretical calculation of the photoionization process. As can be seen, the measured and calculated MFPADs shown in Fig. 2j–l are very similar to the calculated projections in momentum space of the fully correlated ground state wave function shown in Fig. 2g–i. In other words, the momentum of the ejected photoelectron faithfully reflects and maps the momentum of a bound state electron in the molecular ground state when the momentum of the second bound electron is constrained by projection of the $H_2$ wave function onto different molecular-ion states; this represents the correlation between the two electrons. Note in particular Fig. 2g is not empty and Fig. 2h, i are not identical, as it would be for an uncorrelated $H_2$ ground state (compare with Fig. 2a–c).

**Identifying the quantum state of the second electron**. In more detail, the angular emission distributions and the final quantum state of electron B are obtained in our experiment by measuring the momenta of the charged particles generated by the photo-ionization process in coincidence. As the singly charged molecule dissociates in the cases presented here into a neutral H atom and a proton, we can obtain the spatial orientation of the molecular axis by measuring the vector momentum of the proton (i.e., its emission direction after the dissociation). The electron emission direction in the molecular frame is then deduced from the relative emission direction of the proton and the vector momentum of the electron. Additionally, the magnitude of the measured ion momentum provides the kinetic energy release (KER) of the reaction. The latter enables an identification of the quantum state of electron B (i.e., the $H_2^+$ electronic state), which is demonstrated in Fig. 3. Fig. 3a shows the relevant potential energy curves of $H_2^+$ and Fig. 3b the measured (and theoretically predicted) KER spectra. From the measured sum of the kinetic energies of the electron and the proton we furthermore identify the asymptotic electronic state of the neutral H fragment (not detected in the experiment), mostly $H(n = 1)$ and $H(n = 2)$.

**Nodal structure of the wave function**. Our experimentally obtained spectra not only show the imprint of correlation, but also allow us to separate the contribution of different pieces of the electronic wave function to this correlation. Indeed, the momentum distribution of electron A depends strongly on the properties of electron B. The most dramatic example can be seen by comparing the upper and middle rows in Fig. 2, which show electron A under the condition that electron B is detected in the $2p\sigma_u$ and $2s\sigma_g$ states of $H_2^+$, respectively. Upon this change in the selection

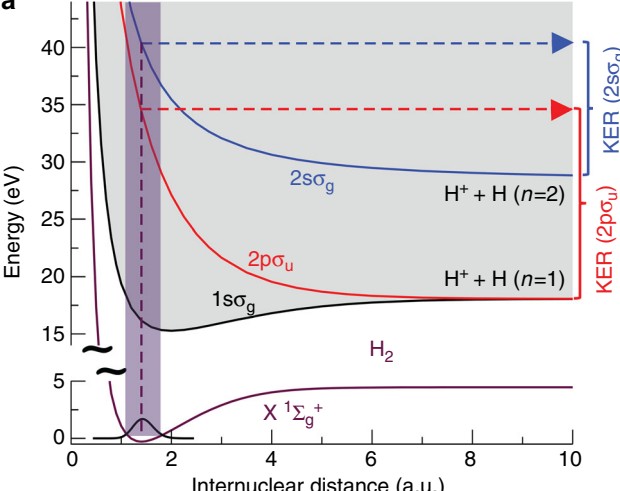

**a**

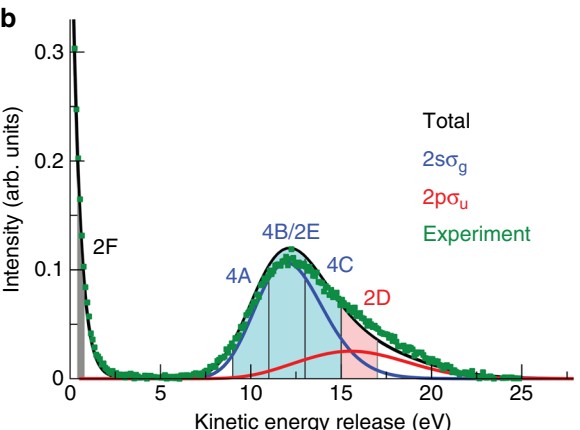

**b**

Fig. 3 Correlation diagram and kinetic energy distribution for dissociation of $H_2^+$ **a** Potential energy curves for the ground state of $H_2$ (lower curve) and the $1s\sigma_g$, $2s\sigma_g$, and $2p\sigma_u$ ionization thresholds (upper curves). The latter correspond to electronic states of $H_2^+$. The violet shaded area represents the Franck-Condon region associated to the ground vibrational state of $H_2$. Notice the break in the energy scale for a better visualization. The dashed violet line shows how the initial internuclear distance of the molecule is mapped onto the kinetic energy release (KER) of the reaction applying the "reflection approximation"[9]. **b** KER distribution obtained after single-photon ionization of $H_2$ employing photons of $h\nu = 400$ eV. Symbols: experiment, lines: theory. The calculation depicted by the black curve includes the twelve states with the highest photo ionization cross sections (up to $n = 4$). The main contributions (besides $1s\sigma_g$ at low KER) are shown in blue ($2s\sigma_g$) and red ($2p\sigma_u$), others are not visible on that scale. The shaded areas indicate the regions of KER selected in Figs. 2d–f and 4a, c

of electron B, the maxima in the momentum distribution of electron A become minima and vice versa. This can be intuitively understood in coordinate space. The maxima in the $k$-space distribution correspond to the constructive interference of the part of the electron density close to one or the other nucleus spaced by $R$. Thus, inverting maxima to minima in $k$-space corresponds to a phase shift of $\pi$ between the wave function at one or the other nucleus in coordinate space. For $H_2$, the two-electron wave function is *gerade*, i.e., it has the same sign of the overall phase at both centers. For a large part of the two-electron wave function, this symmetry consideration is also valid for each individual electron (it reflects the fact that both electrons occupy the $1s\sigma_g$ orbital most of the time). Therefore, both electrons have the same phase at both nuclei, which, in turn, is directly reflected in the

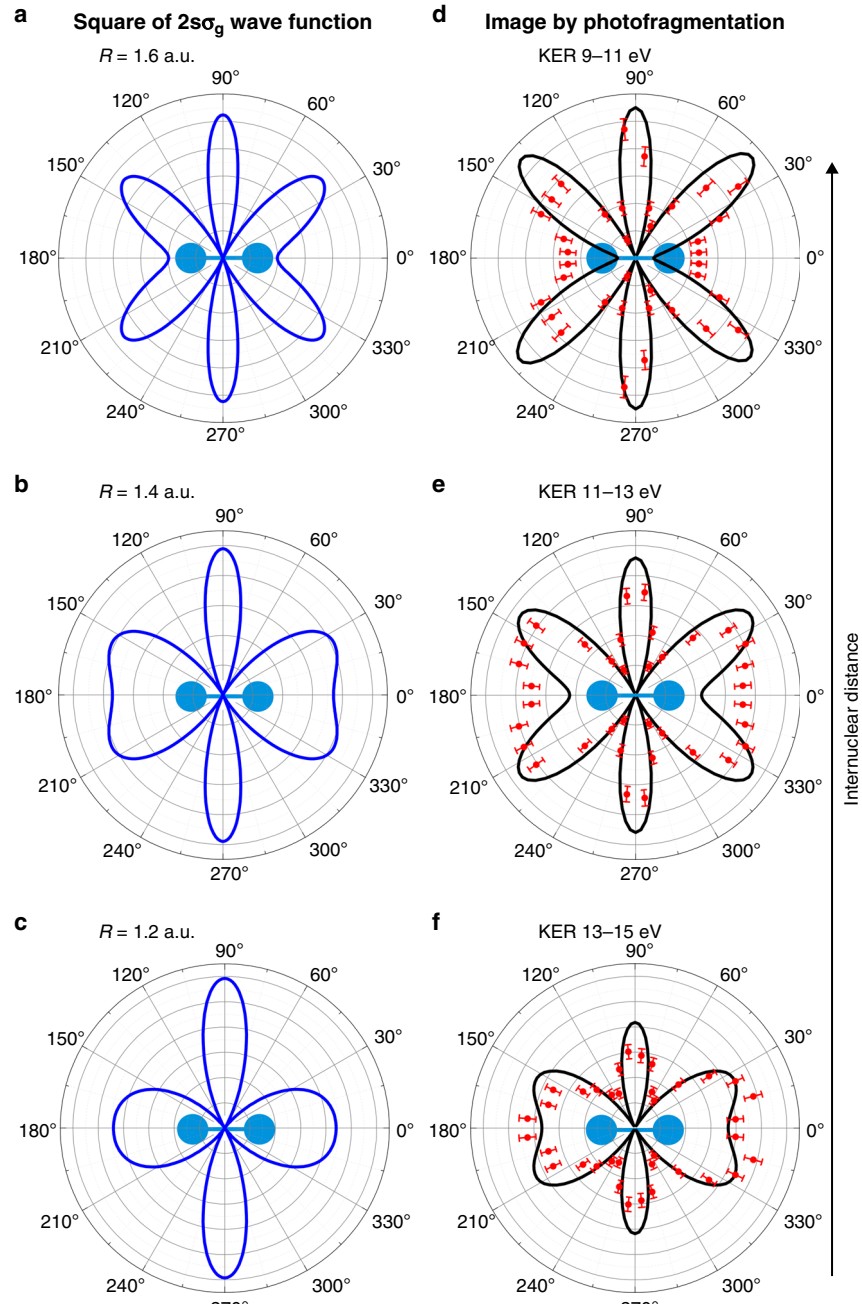

**Fig. 4** Dependence of the momentum distribution on the internuclear distance $R$ **a**–**c** and KER **d**–**f** of the molecule at the instant of photoionization. Molecular orientation as indicated. **a** to **c**: Square of the correlated wave function, as shown in Fig. 2h, but for internuclear distances as stated in the legends. Electron B is projected onto the $2s\sigma_g$ state while electron A is depicted. **d**–**f**: Experimental and theoretical MFPADs (symbols and black line, respectively) for the KER ranges corresponding to the internuclear distances in **a**, **c** resulting from applying the reflection approximation through the $2s\sigma_g$ potential energy curve. The error bars indicate the standard deviation of the mean value

maximum at $k_x = 0$ and the corresponding maximum in the direction perpendicular to the molecular axis in Fig. 2e, f. Due to electron correlation, however, this is not strictly true for all parts of the wave function: Projecting electron B onto the $2p\sigma_u$ state highlights this small fraction of the wave function where electron A has the opposite phase at the two nuclei. As explained before, this part of the wave function does not exist for a Hartree–Fock wave function and Fig. 2a is therefore empty. This phase change of the wave function between the nuclei leads to the nodal line through the center in Fig. 2d and the nodes in Fig. 2g, j in the direction perpendicular to the molecular axis.

In addition to identifying the final state of electron B, the measured KER provides further insights into the ionized $H_2$ molecule. As soon as the potential energy curve relevant for the process is known, one can infer the internuclear distance $R$ of the two atoms of the molecule at the instant of photoabsorption by using the reflection approximation[9] (see Fig. 3). This allows us to investigate more details of the two-electron wave function: The distributions in Fig. 2d–f shows nodal lines that lead to corresponding nodes in the angular distributions in Fig. 2g–i. As mentioned above, these nodes in $k$-space are separated by $\Delta k_x = 2\pi/R$. Within the range of $R$ covered by the

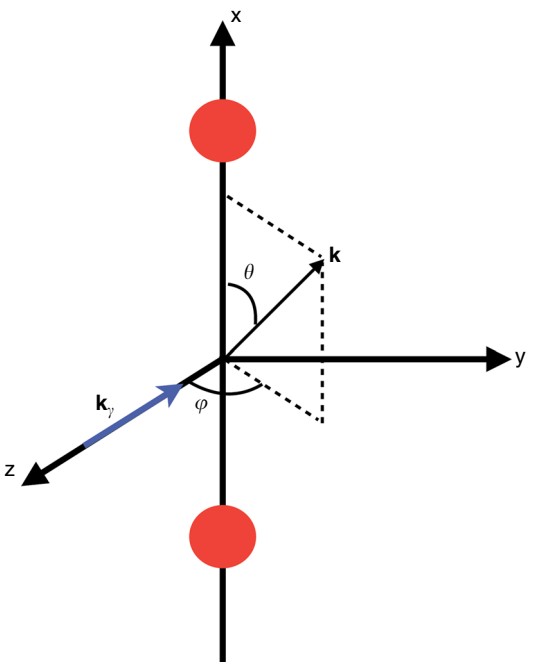

**Fig. 5** Geometrical definitions: polar angle $\theta$ and azimuthal angle $\varphi$ defining the direction of the electron momentum k with respect to the plane defined by the internuclear axis ($x$) and the propagation direction $\mathbf{k}_\gamma$

Franck-Condon region, the nodal structure of the electronic wave function changes significantly and Fig. 4 demonstrates how the $k$-space distribution of the two-electron wave function changes accordingly as a function of $R$ (or KER respectively). The corresponding experimental and theoretical MFPADs resulting from high energy photoionization follow a similar pattern.

In conclusion, high energy angular resolved photoionization is a promising route to access molecular wave functions in momentum space. The process of molecular dissociation in combination with shake up of the bound electron is universal by its nature. Shake up of an electron into a continuum state instead of a bound state, i.e., double ionization of the molecule, might also come into play. Therefore, this approach can in principle be extended to molecules with more than two electrons. In detail, it depends on the shape of the potential energy surfaces which determines to which extend different ionic states can be separated by the kinetic energy of the fragments. Combined with coincidence detection, this technique opens the door to image correlations in electronic wave functions. Similar approaches have also been proposed for imaging correlations in superconductors[10]. With the advent of X-ray free electron lasers and the extension of higher harmonic sources to high photon energies, such correlation imaging bares the promise to make movies of the time evolution of electron correlations in molecules and solid materials.

## Methods

**Experiment.** The experiment has been performed at beamline P04[11] of the PETRA III facility at DESY in Hamburg. The circularly polarized photon beam (400 eV photon energy, about $1.3 \times 10^{13}$ photons per second, 100 μm focus diameter) was crossed with a supersonic $H_2$ gas jet (diameter 1.1 mm, local target density $5 \times 10^{10}$ cm$^{-2}$) at right angle in the center of a COLTRIMS spectrometer[12–14]. A homogeneous electric field of 92 V cm$^{-1}$ guided electrons and ions towards position-sensitive micro-channel plate detectors (active area 80 mm diameter) with hexagonal delayline readout[15]. In the ion arm of the spectrometer a 55 mm acceleration region was followed by a 110 mm drift region. The electron arm of the spectrometer was formed by a 37 mm long acceleration region. A magnetic field of 35.5 G parallel to the electric field guided the electrons on cyclotron trajectories. The data was taken in 480-bunch operation mode, equaling a repetition rate of

62.5 MHz. A residual gas pressure of $2 \times 10^{-10}$ mbar in the reaction chamber led to about 200 Hz of ions detected without the gas jet in operation. The count rates during the experiment were ~ 350 Hz on the ion detector and about 5 kHz on the electron detector. Dissociative ionization events (reaction (5)) were selected by gating on the ion and electron time of flight and on the ion kinetic energy. After all conditions applied to the data, we end up with ~ 200,000 events which we analyze in the MFPADs.

**Correlation imaging.** To first order of perturbation theory, the ionization amplitude of a one-electron molecular system is given by (within the dipole approximation)

$$D = \langle \phi_n(\mathbf{r}) | \hat{\mu}(\mathbf{r}) | \chi_{\mathbf{k}}(\mathbf{r}) \rangle, \tag{6}$$

where $\phi_n$ is the initial state, $\hat{\mu}$ is the dipole operator, and $\chi_{\mathbf{k}}$ is the final state representing a photoelectron with momentum $\mathbf{k}$. At high photoelectron energies, one can approximate the final state by a plane wave, $\chi_{\mathbf{k}}(\mathbf{r}) = e^{i\mathbf{k}\mathbf{r}}$. Thus, if we consider circularly polarized light propagating along the $z$-axis and a molecule fixed along the $x$-axis (see Fig. 5), the transition amplitude can be written, in the velocity gauge:

$$\begin{aligned} D &\simeq \langle \phi_n(\mathbf{r}) | \hat{\mu}(\mathbf{r}) | e^{i\mathbf{k}\mathbf{r}} \rangle \\ &= \langle \phi_n(\mathbf{r}) | \frac{\partial}{\partial x} + i \frac{\partial}{\partial y} | e^{i\mathbf{k}\mathbf{r}} \rangle \\ &= (-k_y + i k_x) \langle \phi_n(\mathbf{r}) | e^{i\mathbf{k}\mathbf{r}} \rangle. \end{aligned} \tag{7}$$

The corresponding photoionization probability (or equivalently the photoionization cross section), differential in the electron emission angles and momentum (or MFPAD), is proportional to the square of the transition amplitude (see Fig. 5 for notations):

$$\frac{dP}{d(\cos\theta)d\varphi dk} \simeq \left(k_x^2 + k_y^2\right) \left|\langle \phi_n(\mathbf{r}) | e^{i\mathbf{k}\mathbf{r}} \rangle\right|^2. \tag{8}$$

Restricting the detection of the electrons to the plane containing the molecule and perpendicular to the light propagation direction, the above expression reduces to:

$$\frac{dP}{d(\cos\theta)dk} = k^2 \left|\langle \phi_n(\mathbf{r}) | e^{i\mathbf{k}\mathbf{r}} \rangle\right|^2. \tag{9}$$

This expression is only applicable to differential probabilities in the $(x, y)$ plane. It can be seen that the integral over $\mathbf{r}$ is proportional to the Fourier transform $\phi_n(\mathbf{k})$ of the $\phi_n(\mathbf{r})$:

$$\frac{dP}{d(\cos\theta)dk} = k^2 (2\pi)^{3/2} \left|\frac{1}{(2\pi)^{3/2}} \int d\mathbf{r} \phi_n(\mathbf{r}) e^{i\mathbf{k}\mathbf{r}}\right|^2, \tag{10}$$

where we have introduced a factor of $(2\pi)^{3/2}$ to make this relationship clearer, i.e.,

$$\frac{dP}{d(\cos\theta)dk} = k^2 (2\pi)^{3/2} |\phi_n(\mathbf{k})|^2. \tag{11}$$

Thus, at high photon energies, the MFPADs measured in the polarization plane of the circularly polarized light directly map the initial electronic wave function.

Let us now generalize this concept to the case of a correlated initial state as that of the $H_2$ molecule. The amplitude describing photoionization from the ground state, $\Psi_0(\mathbf{r}_1, \mathbf{r}_2)$, can now be written as:

$$D = \langle \Psi_0(\mathbf{r}_1, \mathbf{r}_2) | \hat{O}(\mathbf{r}_1, \mathbf{r}_2) | \Phi_f(\mathbf{r}_1, \mathbf{r}_2) \rangle \tag{12}$$

where $\hat{O}(\mathbf{r}_1, \mathbf{r}_2) = \hat{\mu}(\mathbf{r}_1) + \hat{\mu}(\mathbf{r}_2)$ and $\Phi_f(\mathbf{r}_1, \mathbf{r}_2)$ is the final continuum state. At high photoelectron energies, the latter can be approximately written as a product of an $H_2^+$ continuum wave function $\chi_{\mathbf{k}}(\mathbf{r}_2)$ that describes a photoelectron with linear momentum $\mathbf{k}$ and an $H_2^+$ bound wave function $\phi_n(\mathbf{r}_1)$ that describes the electron remaining in the ion:

$$D = \langle \Psi_0(\mathbf{r}_1, \mathbf{r}_2) | \hat{O}(\mathbf{r}_1, \mathbf{r}_2) | \phi_n(\mathbf{r}_1) \chi_{\mathbf{k}}(\mathbf{r}_2) \rangle. \tag{13}$$

We now write the fully correlated ground state wave function of $H_2$ as a linear combination of two-electron configurations expressed as antisymmetrized products of Hartree–Fock (HF) orbitals

$$\begin{aligned} \Psi_0 = & \ 1s\sigma_g^{HF}(\mathbf{r}_1) 1s\sigma_g^{HF}(\mathbf{r}_2) + c_1 2s\sigma_g^{HF}(\mathbf{r}_1) 2s\sigma_g^{HF}(\mathbf{r}_2) \\ & + \ c_2 2p\sigma_u^{HF}(\mathbf{r}_1) 2p\sigma_u^{HF}(\mathbf{r}_2) + ... \end{aligned} \tag{14}$$

where we have factored out the antisymmetric spin wave function corresponding to a singlet multiplicity and $c_1, c_2 \ll 1$. The first term in this expansion represents the ground state of $H_2$ in the HF approximation,

$$\Psi_0^{HF}(H_2) = 1s\sigma_g^{HF}(\mathbf{r}_1) 1s\sigma_g^{HF}(\mathbf{r}_2), \tag{15}$$

which includes screening and exchange but neglects electron correlation. Substituting Eq. (14) in Eq. (13), retaining the lowest-order non-zero terms, and

using Eq. (11), the partial differential photoionization cross sections (or partial MFPADs) associated with the lowest three ionization channels, $1s\sigma_g$, $2s\sigma_g$, and $2p\sigma_u$, can be written (up to a trivial factor of $k^2(2\pi)^{3/2}$):

$$\left|\langle\Psi_0|\hat{O}|1s\sigma_g\chi_{\mathbf{k}}\rangle\right|^2 \simeq \left|\langle 1s\sigma_g^{HF}|1s\sigma_g\rangle\right|^2\left|\phi_{1s\sigma_g^{HF}}(\mathbf{k})\right|^2, \tag{16}$$

$$\left|\langle\Psi_0|\hat{O}|2s\sigma_g\chi_{\mathbf{k}}\rangle\right|^2 \simeq \left|\langle 1s\sigma_g^{HF}|2s\sigma_g\rangle\right|^2\left|\phi_{1s\sigma_g^{HF}}(\mathbf{k})\right|^2, \tag{17}$$

$$\left|\langle\Psi_0|\hat{O}|2p\sigma_u\chi_{\mathbf{k}}\rangle\right|^2 \simeq c_2\left|\langle 2p\sigma_u^{HF}|2p\sigma_u\rangle\right|^2\left|\phi_{2p\sigma_u^{HF}}(\mathbf{k})\right|^2, \tag{18}$$

where the dependence on $\mathbf{r}_1$ and $\mathbf{r}_2$ is now implicit in all equations. Hence, the partial differential cross sections are proportional to the representation of the ground state HF orbitals in momentum space and to the overlap between these HF orbitals and the $H_2^+$ orbitals that define the different ionization thresholds. As can be seen, in the absence of electron correlation, i.e., when the initial state is simply described by $\Psi_0^{HF}$ and therefore the $c_i$ coefficients are zero, ionization can be direct (i.e., an electron is ejected into the continuum and the other remains in the $1s\sigma_g$ orbital, Eq. (16) or can be accompanied by excitation of the remaining electron into the $2s\sigma_g$ state (shake-up mechanism, Eq. (17)). Ionization and excitation into the $2p\sigma_u$ state is only possible when $c_2$ is different from zero (Eq. (18)), i.e., when electron correlation is not negligible.

To get additional information about the relative magnitude of the partial cross sections, we write the HF orbitals as linear combinations of $H_2^+$ orbitals. To the first order of perturbation theory,

$$1s\sigma_g^{HF} = 1s\sigma_g + \lambda_1 2s\sigma_g + \dots \tag{19}$$

$$2p\sigma_u^{HF} = 2p\sigma_u + \lambda_2 3p\sigma_u + \dots \tag{20}$$

and so on, where $\lambda_i \ll 1$. Substituting Eqs. (19) and (20) in (16), (17), and (18), and retaining the lowest-order non-zero terms in $\lambda_i$, one obtains the following simplified expressions for the three ionization channels above:

$$\left|\langle\Psi_0|\hat{O}|1s\sigma_g\chi_{\mathbf{k}}\rangle\right|^2 \simeq \left|\phi_{1s\sigma_g}(\mathbf{k})\right|^2 \tag{21}$$

$$\left|\langle\Psi_0|\hat{O}|2s\sigma_g\chi_{\mathbf{k}}\rangle\right|^2 \simeq \lambda_1\left|\phi_{1s\sigma_g}(\mathbf{k})\right|^2, \tag{22}$$

$$\left|\langle\Psi_0|\hat{O}|2p\sigma_u\chi_{\mathbf{k}}\rangle\right|^2 \simeq c_2\left|\phi_{2p\sigma_u}(\mathbf{k})\right|^2, \tag{23}$$

where we have used the fact that the $H_2^+$ orbitals form an orthonormal basis. As can be seen, the dominant mechanism is direct ionization from the $1s\sigma_g$ orbital (Eq. (21)). Ionization with simultaneous excitation of the remaining electron (Eqs. (22) and (23)) is much less likely, since both $\lambda_1$ and $c_2$ are small. Ionization through other channels only contribute to second or higher order, thus explaining why they barely contribute to the ionization cross section. According to this simple formalism, for both the $1s\sigma_g$ and $2s\sigma_g$ channels, the MFPADs map the $1s\sigma_g$ orbitals in momentum space. The only difference between them is the absolute value of the electron momentum (or electron kinetic energy) used to perform the mapping. In contrast, the MFPAD for the $2p\sigma_u$ channel maps the $2p\sigma_u$ orbital in momentum space. As explained in the text, these three channels lead to dissociative ionization in different KER regions: $1s\sigma_g$ mainly contributes at low KER, $2s\sigma_g$ at intermediate KER and $2p\sigma_u$ at high KER. Therefore, the analysis of the MFPADs in different KER regions provides information about the three different mechanisms: direct ionization, shake-up ionization and ionization driven by electron correlation.

Testing electron correlation by one photon two electron processes has a long history (see, e.g., refs. [16,17] for early proposals). Previous works often focused on the probability of double ionization (see ref. [18] for a review) or angular distributions for double ionization of molecules (see, e.g., ref. [19]). The present work differs from these earlier ones by the high electron energy which allows for a direct interpretation of the angular distribution as being an image of the ground state wave function (plane wave or Born approximation). In contrast, at lower electron energies, as they were used in previous works, the electron angular distributions are shaped by the subtle interplay between three effects: electron correlation in the initial state, scattering correlations during the ionization process[20,21] and the ionic potential.

It is worth noticing that the specific form of the MFPADs resulting from using Eqs. (16), (17), and (18) (or their simplified versions (21)–(23)) is the consequence of the dipole selection rule that operates in this particular problem. As a consequence, for transition operators $\hat{O}$ different from the dipole one, different expressions would be obtained. Nevertheless, even in this case, one can anticipate that in the absence of electron correlation, the matrix elements given by Eqs. (17) and (18) (or equivalently (22) and (23)) would be strictly zero.

**Ab initio calculations**. The ab initio method used to obtain the dissociative ionization spectra and the corresponding angular distributions has been described elsewhere[22,23]. It has been successfully applied to evaluate photoionization cross sections and MFPADs of the $H_2$ molecule in both time-dependent and time-independent scenarios[22-25]. Due to the high photoelectron energies produced in the experiment, we have made use of the Born-Oppenheimer approximation, which allows us to describe the initial and final continuum wave functions as products of an electronic wave function and a nuclear wave function. The ground state electronic wave function has been obtained by performing a configuration interaction calculation in a basis of antisymmetrized products of one-electron $H_2^+$ orbitals, and the final electronic continuum states by solving the multichannel scattering equations in a basis of uncoupled continuum states that are written as products of a one-electron wave function for the bound electron and an expansion on spherical harmonics and B-spline functions for the continuum electron. The multichannel expansion includes the six lowest ionic states ($1s\sigma_g$, $2p\sigma_u$, $2p\pi_u$, $2s\sigma_g$, $3d\sigma_g$, and $3p\sigma_u$) and partial waves for the emitted electron up to a maximum angular momentum $l_{max} = 7$ enclosed in a box of 60 a.u., which amounts to around 61,000 discretized continuum states. The one-electron orbitals for the bound electron are consistently computed in the same radial box using single-center expansions with corresponding angular momenta up to $l_{max} = 16$. The electronic wave functions have been calculated in a dense grid of internuclear distances comprised in the interval $R = [0, 12]$ a.u. The nuclear wave functions have been obtained by diagonalizing the corresponding nuclear Hamiltonians in a basis of B-splines within a box of 12 a.u. We have thus computed the photoionization amplitudes and cross sections for circularly polarized light for the dissociative ionization process from first order perturbation theory

$$D = \langle\Phi_f(\mathbf{r}_1,\mathbf{r}_2)\xi_f(R)|\hat{O}(\mathbf{r}_1,\mathbf{r}_2)|\Psi_0(\mathbf{r}_1,\mathbf{r}_2)\xi_0(R)\rangle. \tag{24}$$

At variance with Eq. (12), the previous equation includes the initial $\xi_0$ and final $\xi_f$ vibrational wave functions and integration is performed over both electronic and nuclear coordinates.

The present methodology does not account for the double ionization channel, which is open at the photon energies used in the present work. However, this channel is expected to have a marginal influence in the reported results since the corresponding cross section is at least an order of magnitude smaller than that for the single ionization channel. In addition, according to the Franck–Condon picture, double ionization could only contribute to the KER spectrum in the region around 19–20 eV, i.e., outside the region of interest discussed in the present work.

**Data availability**. The data that support the findings of this study are available from the authors on reasonable request.

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

## Acknowledgements

This work was funded by the Deutsche Forschungsgemeinschaft, the BMBF, the European Research Council under the European Union Seventh Framework Programme (FP7/2007-2013)/ERC grant agreement 290853 XCHEM, the MINECO projects FIS2013-42002-R and FIS2016-77889-R, and the European COST Action XLIC CM1204. All calculations were performed at the CCC-UAM and Mare Nostrum Supercomputer Centers. We are grateful to the staff of PETRA III for excellent support during the beam time. K.M. and M.M. would like to thank the DFG for support via SFB925/A3. A.K. and V.S. thank the Wilhelm und Else Heraeus-Foundation for support. J.L. would like to thank the DFG for support. S.K. acknowledges support from the European Cluster of Advanced Laser Light Sources (EUCALL) project which has received funding from the European Union's Horizon 2020 research and innovation programme under grant agreement No 654220. T.W. was supported by the U.S. Department of Energy Basic Energy Sciences under Contract No. DE-AC02-05CH11231. A.P. acknowledges a Ramón y Cajal contract from the Ministerio de Economa y Competitividad. We thank M. Lara-Astiaso for providing us with the Hartree–Fock wave functions.

## Author contributions

M.W., D.M., J.L., F.T., C.S., M.K., M.P., K.M., M.M., J.V., S.K., L.Ph.H.S., J.B.W., M.S.S., T.J., and R.D. contributed to the experiment. R.Y.B., V.V.S., A.S.K., L.A., A.P. and F.M. performed the calculations. M.W., R.Y.B., D.M., J.L., F.T., C.S., M.K., U.L., M.P., K.M., M.M., J.V., S.K., T.W., L.Ph. H.S., J.B.W., M.S.S., V.V.S., A.S.K., L.A., A.P., F.M., T.J., and R.D. contributed to the manuscript. M.W. and R.D. did the data analysis.

## Additional information

**Competing interests:** The authors declare no competing financial interests.

