## [Peer Review File · Nature Communications]

Reviewers' comments:

Reviewer #1 (Remarks to the Author):

This is an original and highly interesting work on the important aspect of imaging the correlated two-electron wave function as exemplified here for a fundamental two-electron system, the hydrogen molecule. The work comprises both cutting-edge experimental results and a very thorough theoretical analysis which fit very well together. The work is very well structured and clearly formulated which makes it easy accessible also for a non-specialised readers. Furthermore, I can clearly foresee that the work presented has a high potential of influencing and stimulating new thinking in the broad areas of physics and chemistry. Therefore, I do warmly recommend this milestone work for publication without any major changes.

In view of future investigations building on the present work, I wonder if the authors could possibly add on some hints on how their approach presented can be applied to other multi-electron systems which consists of more than two electrons? Will it be straight forward or are there any major challenges which the authors can foresee already now?

A second aspect which the authors could expand on concerns the experimental part of the Methods section. In order to assess a typical experimental run time, I wonder if the authors could possibly share with us some on the actual count rate used and if they applied specified strategies in down-selecting the coincidence events sought for.

Reviewer #2 (Remarks to the Author):

In this theory-experiment joint work, authors studied the single-photon ionization of H₂ in XUV fields. By coincidentally measuring the photoelectron momentum distribution and the KER of the dissociative fragments, one may image the two-electron correlated wave function. This experiment is novel, and this imaging method can be generalized to many atoms and molecules. I recommend to publish it if following questions can be addressed.

(1) Figs.2 B, C, D, E, F show the bound electron momentum distributions of H₂⁺. Though others look reasonable, I have no idea why there is a circle around the zero point in Fig. 1(c).

(2) G-I and J-L in Fig. 2 show the photoelectron angular distributions. While G-I is the accurate results after counting on the electron correlation, I didn't get how the theoretical calculations performed for J-L. Why does it call "nearly exact theoretical calculation"? Is that the perturbative calculation explained in the method section? This question also goes to Fig. 4 left and right columns.

(3) In Fig. 2, panels B, E and panels C, D, F are performed for $k=5.3$ and 5.2 a.u.. Why is it necessary to use different photon energies?

(4) At the end of the "Method: correlation imaging", authors stated that the three KER regions correspond to three mechanisms: direct, shake-up and the ionization driven by electron correlation.

Please explain the fundamental difference between shakeup ionization and the ionization driven by electron correlation.

Some other minor points:

(5) In Eq. (7), the laser vector potential probably is missing.

(6) In the sentence "...is ejected into the continuum and the other remains in the $1s\sigma_g$ orbital, eq. (16)", the ")" is missing.

Reviewer #3 (Remarks to the Author):

The authors of "Imaging the correlated two-electron wave function of a hydrogen molecule" have carried out an impressive set of experiments and calculations investigating electron correlations in molecules, a truly complex and important subject. The authors found a set of experimental and theoretical conditions which allows for determining the final state of an unmeasured electron by detecting an ionized electron and a proton from the dissociative ionization of H₂. I think this work is both fascinating and of general importance, and should be considered for publication in Nature Communications. However, I have a few comments which the authors might want to consider:

1) The overall text is very clearly written, I would like to point out on page 2, just after equation 1, the use of the word "target" is unclear. What are the "target" electrons?

2) In the same paragraph as the previous point, the use of the word "simply" is distracting. While experts might think differential cross sections and Fourier transforms are "simple," I would guess the general reader might be discouraged by this language.

3) In Fig. 2 caption, one $|k|$ has an arrow, and one does not. Also, I am assuming in panels J-L that the symbols are the experimental data while the line is theoretical.

4) I would be very careful in saying that you are imaging a wave function (i.e. in the title).

5) A concise description of what is left out of the otherwise near-exact calculations would be nice.

6) Restrictions to the method and/or a short description of its general applicability, especially regarding the sentence on page 4 which starts with "In other words, the momentum of the ejected photoelectron faithfully reflects and maps the momentum of a bound state electron in the molecular ground state when..." seems to be missing. Is this approach limited to diatomic molecules with only 2 electrons?

7) Is there double ionization? If yes, how do you exclude these events?

Reviewer #4 (Remarks to the Author):

The major claim of this work is to demonstrate an imaging technique for direct observation of the correlation between electrons in wave-functions describing molecular bound states. It addresses here imaging of the correlated two-electron wave function of the H₂ ground-state.

The method consists of measuring molecular frame photoelectron angular distributions (MFPADs) for one-photon high energy single ionization of the H₂ molecule into well defined H₂⁺(n λ) ionic states, in

specific conditions, i.e. using circularly polarized light and recording electrons emitted in the light polarization plane for molecules lying in that plane. The experimental technique relies on coincident detection of the electronic and ionic fragments produced in dissociative photoionization of H₂ and determination of their momenta, in a photon energy range of few hundreds of eV. Ab initio calculations of the MFPADs at similar energies, together with that of selected components of the correlated wave function, support the experimental results. The work presented establishes that the achieved MFPADs provide a good representation of the one-electron momentum distributions resulting from the projection of the correlated two-electron ground state wave function onto the lowest electronic states of H₂⁺, and also addresses their dependence on the internuclear distance.

Imaging directly different parts of many-electron wave functions which specifically characterize electronic correlation in molecular bound states, as reported here by M. Waitz et al, is novel to my knowledge and the authors present a set of original experimental and theoretical results of high quality for the two-electron H₂ molecule, supported by a detailed analysis.

Although the state of the art experimental and theoretical methods implemented for this photoionization study are well identified, and MFPADs observables are known to be highly sensitive to the photoionization dynamics including electronic correlation, the novelty of the work presented lies in their successful application in appropriately chosen kinematic conditions which exhibit the electronic correlation in the initial neutral state of the ionized molecule. Measuring MFPADs in such conditions, where the continuum electron of few hundreds of eV energy can be described by a plane wave, is quite challenging.

The reported results are clear, convincing and of great interest for the field addressing fundamental processes in molecular physics. The manuscript is well organized and written.

Following comments may be considered by the authors.

In my opinion, it would be of interest that the authors position briefly the present work in the context of related studies, such as single-photon double ionization of H₂ which is somehow an extension of single photoionization with excitation of the bound electron, with similar problematics addressing the highly correlated initial-state wave function. Several papers were reported in particular by the Frankfurt group and co-workers, using the COLTRIMS technique in a similar instrumental configuration as discussed here, therefore it sounds relevant.

The main results of the present work are reported in Figure 2, while the conditions for their achievement relies on Figure 3 A and B. The H₂⁺ ionic states and potential energy curves presented, and accordingly the related discussion, focus on the three lowest states of σ symmetry, ignoring the possible contribution of e.g. the 2p_{ru} ionic channel (included in the ab initio calculations) which dissociates into the H⁺ + H(n=2) channel and could probably not be resolved from the 2s σ g channel experimentally: ionization to both channels is observed at lower energies (e.g. Ito et al J. Phys. B 2000). The restriction to the three lowest states should be justified.

On the other hand one may wonder if double ionization, which leads to a fast electron and a slow electron as such photon energies, could also partially contribute to the (H⁺,e) coincident events at the highest KERs considered here (14-22 eV): is such a contribution excluded here ?

The discussion of the results in Figure 2 clearly supports the developments given in the Supplementary Material, which provide the expression of the partial MFPADs associated with the assigned ionization channels in terms of the ground state HF orbitals representation in momentum space. They illustrate the statement (end of the first column of page 4) "It is solely due to electron correlation that the momentum distribution of electron A depends strongly on the properties of electron B".

This statement can also be thought from the photoionization perspective. If indeed it is solely due to

electron correlation that channels involving ionization and excitation are populated, the $g \rightarrow u$ dipole selection rule plays a role in imposing an interdependence between the characteristics of electron A and electron B. The continuum molecular state being of u symmetry, the electronic wave function of electron A should be of g or u symmetry under the condition that electron B is detected in the $2p\sigma_u$ or $2s\sigma_g$ state of H_2^+ , respectively ? This has consequences in terms of the different partial waves (with defined node properties) which contribute to the momentum distribution of electron A and it holds for the both the parallel and perpendicular transitions which add up in the observed MFPAD. Could the authors comment briefly on this perspective, which is not so clear from eq (16-18) ?

In the discussion of Figure 4 describing the dependence of the momentum distributions and MFPADs with the internuclear distance (and KER through the reflection approximation), in addition to the similar trend found between the A to C momentum distributions and the D to F MFPADs, it is remarkable that looking at the computed MFPADs the highest resemblance is between A and E, and B and F. Any comment to be added in the text ?

Methods

In the part devoted to the experiment it would be interesting that the authors provide more detail on the conditions enabling collection and momentum determination of the 360-380 eV electrons, at the core of this work: little is said about the electron arm of the spectrometer used and the cited references do not address explicitly such high energy conditions. The related resolution in momentum determination, and therefore in energy and angle, is of importance for the discussion of the measured MFPADs.

For a proper use of the count rate numbers given in this section (200 Hz of ions without the gas jet and 350 Hz with the gas jet), one should know which ions do the authors refer to ? are these numbers given for the H^+ signal or for all ions ?

Below, we list our responses to the points raised by the reviewers:

- **Response to reviewer #1:**

Reviewer:

This is an original and highly interesting work on the important aspect of imaging the correlated two-electron wave function as exemplified here for a fundamental two-electron system, the hydrogen molecule. The work comprises both cutting-edge experimental results and a very thorough theoretical analysis which fit very well together. The work is very well structured and clearly formulated which makes it easy accessible also for a non-specialised readers. Furthermore, I can clearly foresee that the work presented has a high potential of influencing and stimulating new thinking in the broad areas of physics and chemistry. Therefore, I do warmly recommend this milestone work for publication without any major changes.

In view of future investigations building on the present work, I wonder if the authors could possibly add on some hints on how their approach presented can be applied to other multi-electron systems which consists of more than two electrons? Will it be straight forward or are there any major challenges which the authors can foresee already now?

Our Response:

We added the passage,

“The process of molecular dissociation in combination with shake up of the bound electron is universal by its nature. Shake off of an electron into a continuum state instead of a bound state, i.e. double ionization of the molecule, might also come into play. Therefore, this approach can in principle be extended to molecules with more than two electrons. In detail, it depends on the shape of the potential energy surfaces which determines to which extend different ionic states can be separated by the kinetic energy of the fragments.”

to the outlook section of the manuscript.

Reviewer:

A second aspect which the authors could expand on concerns the experimental part of the Methods section. In order to assess a typical experimental run time, I wonder if the authors could possibly share with us some on the actual count rate used and if they applied specified strategies in down-selecting the coincidence events sought for.

Our Response:

We inserted the following sentences in the Methods section:

“The data was taken in 480-bunch operation mode, equaling a repetition rate of 62.5 MHz. A residual gas pressure of $2 \cdot 10^{-10}$ mbar in the reaction chamber led to about 200 Hz of ions detected without the gas jet in operation. The count rates during the experiment were approx. 350 Hz on the ion detector and about 5 kHz on the electron detector. Dissociative ionization events (reaction (5)) were selected by gating on the ion and electron time of flight and on the ion kinetic energy. After all conditions applied to the data, we end up with approx. 200,000 events which

we analyze in the MFPADs.”

- **Response to reviewer #2:**

Reviewer:

In this theory-experiment joint work, authors studied the single-photon ionization of H₂ in XUV fields. By coincidentally measuring the photoelectron momentum distribution and the KER of the dissociative fragments, one may image the two-electron correlated wave function. This experiment is novel, and this imaging method can be generalized to many atoms and molecules. I recommend to publish it if following questions can be addressed.

(1) Figs.2 B, C, D, E, F show the bound electron momentum distributions of H₂⁺. Though others look reasonable, I have no idea why there is a circle around the zero point in Fig. 1(c).

Our Response:

The circles in Figs. 2 B - F as well as the one in Fig. 1 C indicate the momentum of an electron ionized by a 400 eV photon (which equals 5.2 and 5.3 a.u., respectively). Our intention is to guide the readers eye and make it easier to connect the nodes in the polar plots in Fig. 2 G-I and 1 D, respectively, to the vertical nodal lines in the two dimensional plots mentioned above.

We tried to clarify this with the sentence

“The dashed line indicates the region of momentum space associated with an electron kinetic energy of 380 eV (i.e., a radius of $k = 5.3$ a.u.)”

in the caption of figure 1.

Reviewer:

(2) G-I and J-L in Fig. 2 show the photoelectron angular distributions. While G-I is the accurate results after counting on the electron correlation, I didn't get how the theoretical calculations performed for J-L. Why does it call “nearly exact theoretical calculation”? Is that the perturbative calculation explained in the method section? This question also goes to Fig. 4 left and right columns.

Our Response:

This is very closely linked to the question above. The panels in the third column of Fig. 2 (G - I) show a polar plot of the electron density in its initial bound state (in momentum space, for a fully correlated wave function), i.e. the intensity added up along the circles in the respective panels in the second column (D - F). So the third column is also under the headline “wave function”, as indicated in the figure. The fourth column, in contrast, shows the results of the theoretical calculations for photoionization (the one described in the methods section), together with the experimental results.

The same applies for Fig. 4, the left column shows the square of the wave function, the right column shows the results of the calculations for photoionization together with the experimental results (as indicated in the figure).

To make this point clearer, we changed the caption of figure 2 to

“G - I: ground state wave function (intensity distributions along the circular lines

shown in panels D - F)”

Reviewer:

(3) In Fig. 2, panels B, E and panels C, D, F are performed for $k = 5.3$ and 5.2 a.u.. Why is it necessary to use different photon energies?

Our Response:

The panels are for the same photon energy (400 eV). However, the electron energy is different, as for the three different states excitation requires some of the energy of the photon. Hence, the energy of the emitted electron changes slightly. The calculated momenta for these two cases are 5.2 a.u. ($2s\sigma_g$) and 5.3 a.u. ($1s\sigma_g$ and $2p\sigma_u$), respectively.

Reviewer:

(4) At the end of the “Method: correlation imaging”, authors stated that the three KER regions correspond to three mechanisms: direct, shake-up and the ionization driven by electron correlation. Please explain the fundamental difference between shakeup ionization and the ionization driven by electron correlation.

Our Response:

Direct ionization refers to a process where the bound electron is not excited but remains in the lowest possible state ($1s\sigma_g$ in this case). In contrast to that, the second electron can also be shaken up to an excited state which can be described within the Hartree- Fock approximation (here, $2p\sigma_u$). The third possibility refers to a process where the bound electron occupies an excited state which is not part of the basis used in the Hartree-Fock approximation ($2s\sigma_g$), driven by electron correlation. We tried to explain this vocabulary in the paragraph following eq. (18) in the correlation imaging section. It reads,

“... ionization can be direct (i.e., an electron is ejected into the continuum and the other remains in the $1s\sigma_g$ orbital, eq. (16)), or can be accompanied by excitation of the remaining electron into the ($2s\sigma_g$) state (shake-up mechanism, eq. (17)). Ionization and excitation into the $2p\sigma_u$ state is only possible when c_2 is different from zero (eq. (18)), i.e., when electron correlation is not negligible.”

Reviewer:

Some other minor points:

(5) In Eq. (7), the laser vector potential probably is missing.

Our Response:

The vector potential is included in equations (6) and (7). μ is the dipole operator in first order perturbation theory within the dipole approximation, being a result of the vector potential in velocity gauge.

Reviewer:

(6) In the sentence “...is ejected into the continuum and the other remains in the $1s\sigma_g$ orbital, eq. (16)”, the “)” is missing.

Our Response:

The missing bracket has been added.

- **Response to reviewer #3:**

Reviewer:

The authors of "Imaging the correlated two-electron wave function of a hydrogen molecule" have carried out an impressive set of experiments and calculations investigating electron correlations in molecules, a truly complex and important subject. The authors found a set of experimental and theoretical conditions which allows for determining the final state of an unmeasured electron by detecting an ionized electron and a proton from the dissociative ionization of H₂. I think this work is both fascinating and of general importance, and should be considered for publication in Nature Communications. However, I have a few comments which the authors might want to consider:

1) The overall text is very clearly written, I would like to point out on page 2, just after equation 1, the use of the word "target" is unclear. What are the "target" electrons?

Our Response:

The word *target* is indeed misleading, we deleted it.

Reviewer:

2) In the same paragraph as the previous point, the use of the word "simply" is distracting. While experts might think differential cross sections and Fourier transforms are "simple," I would guess the general reader might be discouraged by this language. **Our Response:**

We agree that the use of the word "simply" is not adequate here and removed it from the text.

Reviewer:

3) In Fig. 2 caption, one $|k\rangle$ has an arrow, and one does not. Also, I am assuming in panels J-L that the symbols are the experimental data while the line is theoretical.

Our Response:

We added the missing vector arrow.

Indeed, the symbols are the data and the green line is theory. We added

"Experimental and theoretical MFPADs (symbols and green line, respectively) ..."

to the caption. We did the same for the caption of figure 4, where the situation is similar.

Reviewer:

4) I would be very careful in saying that you are imaging a wave function (i.e. in the title).

Our Response:

The Reviewer is correct, we image the square of the wave function, not its phase. We leave it up to the editors if the title should be changed to

"Imaging the square of the correlated two-electron wave function of a hydrogen molecule"

Which would be more precise.

Reviewer:

5) A concise description of what is left out of the otherwise near-exact calculations would be nice.

Our Response:

The main ingredient left out in our calculations is the double ionization channel. However, the corresponding cross section is at least an order of magnitude smaller than that for the single ionization channel investigated in the manuscript. Furthermore, based on the Franck-Condon approximation, one can anticipate that the double ionization channel could only contribute to the KER spectrum in the region around 19 eV, i.e., outside the region of interest discussed in the present work. Therefore, we do not expect that inclusion this channel has any influence on the reported results and discussion (as illustrated by the very good agreement between the calculated and measured KER spectrum shown in Fig. 3 B). In any case, we have followed the reviewer's advice and included a sentence in the Supplementary Material section to explicitly mention that the double ionization channel is not included in the present calculations.

We included the following sentences at the end of the Supplementary Material:

"The present methodology does not account for the double ionization channel, which is open at the photon energies used in the present work. However, this channel is expected to have a marginal influence in the reported results since (i) the corresponding cross section is at least an order of magnitude smaller than that for the single ionization channel and (ii) according to the Franck-Condon picture, double ionization could only contribute to the KER spectrum in the region around 19-20 eV, i.e., outside the region of interest discussed in the present work."

Reviewer:

6) Restrictions to the method and/or a short description of its general applicability, especially regarding the sentence on page 4 which starts with "In other words, the momentum of the ejected photoelectron faithfully reflects and maps the momentum of a bound state electron in the molecular ground state when..." seems to be missing. Is this approach limited to diatomic molecules with only 2 electrons?

Our Response:

This is closely connected to the first question of Reviewer #1. We added the passage,

"The process of molecular dissociation in combination with shake up of the bound electron is universal by its nature. Shake up of an electron into a continuum state instead of a bound state, i.e. double ionization of the molecule, might also come into play. Therefore, this approach can in principle be extended to molecules with more than two electrons. In detail, it depends on the shape of the potential energy surfaces which determines to which extend different ionic states can be separated by the kinetic energy of the fragments."
to the outlook section of the manuscript.

Reviewer:

7) Is there double ionization? If yes, how do you exclude these events?

Our Response:

Yes, we measure double ionization events at the same time.

We exclude them from the dataset by making use of a photo ion photo ion

coincidence map on the one hand and by exploiting momentum conservation on the other hand. For those double ionization events where we fail to detect the second proton due to detector efficiency we estimated and subtracted the resulting background. Figure A visualizes the resulting difference in the KER spectrum (curves are normalized to their integral):

Figure A: KER spectrum with and without background subtraction resulting from false coincidences triggered by double ionization events.

- **Response to reviewer #4:**

Reviewer:

The major claim of this work is to demonstrate an imaging technique for direct observation of the correlation between electrons in wave-functions describing molecular bound states. It addresses here imaging of the correlated two-electron wave function of the H₂ ground-state.

The method consists of measuring molecular frame photoelectron angular distributions (MFPADs) for one-photon high energy single ionization of the H₂ molecule into well defined H₂⁺(n,l λ) ionic states, in specific conditions, i.e. using circularly polarized light and recording electrons emitted in the light polarization plane for molecules lying in that plane. The experimental technique relies on coincident detection of the electronic and ionic fragments produced in dissociative photoionization of H₂ and determination of their momenta, in a photon energy range of few hundreds of eV. Ab initio calculations of the MFPADs at similar energies, together with that of selected components of the correlated wave function, support the experimental results. The work presented establishes that the achieved MFPADs provide a good representation of the one- electron momentum distributions resulting from the projection of the correlated two- electron ground state wave function onto the lowest electronic states of H₂⁺, and also addresses their dependence on the internuclear distance.

Imaging directly different parts of many-electron wave functions which specifically characterize electronic correlation in molecular bound states, as reported here by M. Waitz et al, is novel to my knowledge and the authors present a set of original experimental and theoretical results of high quality for the two-electron H₂ molecule, supported by a detailed analysis.

Although the state of the art experimental and theoretical methods implemented for this photoionization study are well identified, and MFPADs observables are known to be highly sensitive to the photoionization dynamics including electronic correlation, the novelty of the work presented lies in their successful application in appropriately chosen kinematic conditions which exhibit the electronic correlation in the initial neutral state of the ionized molecule. Measuring MFPADs in such conditions, where the continuum electron of few hundreds of eV energy can be described by a plane wave, is quite challenging.

The reported results are clear, convincing and of great interest for the field addressing fundamental processes in molecular physics. The manuscript is well organized and written.

Following comments may be considered by the authors.

In my opinion, it would be of interest that the authors position briefly the present work in the context of related studies, such as single-photon double ionization of H₂ which is somehow an extension of single photoionization with excitation of the bound electron, with similar problematics addressing the highly correlated initial-state wave

function. Several papers were reported in particular by the Frankfurt group and co-workers, using the COLTRIMS technique in a similar instrumental configuration as discussed here, therefore it sounds relevant.

Our Response:

We inserted the following brief discussion at the end of the subsection “Correlation Imaging” of the methods section:

“Testing electron correlation by one photon two electron processes has a long history (see e.g. [19, 20] for early proposals). Previous works often focused on the probability of double ionization (see [21] for a review) or angular distributions for double ionization of molecules (see e.g. [22]). The present work differs from these earlier ones by the high electron energy which allows for a direct interpretation of the angular distribution as being an image of the ground state wave function (plane wave or Born approximation). In contrast, at lower electron energies, as they were used in previous works, the electron angular distributions are shaped by the subtle interplay between three effects: electron correlation in the initial state, scattering correlations during the ionization process ([23, 24]) and the ionic potential.”

Reviewer:

The main results of the present work are reported in Figure 2, while the conditions for their achievement relies on Figure 3 A and B. The H₂⁺ ionic states and potential energy curves presented, and accordingly the related discussion, focus on the three lowest states of σ symmetry, ignoring the possible contribution of e.g. the $2p\pi_u$ ionic channel (included in the ab initio calculations) which dissociates into the H⁺ + H(n=2) channel and could probably not be resolved from the $2s\sigma_g$ channel experimentally: ionization to both channels is observed at lower energies (e.g. Ito et al J. Phys. B 2000). The restriction to the three lowest states should be justified.

Our Response:

The theoretical calculations include the first 12 states with the highest photo ionization cross section, ranging up to $n = 4$. The calculations show that, other than in the low photon energy regime (as e.g. in the work by Ito et al quoted by the reviewer), the influence of other states besides the three ones mentioned, is negligible on the scale of figure 3B.

We added the sentences,

“The calculation depicted by the black curve includes the twelve states with the highest photo ionization cross sections (up to $n = 4$). The main contributions (besides $1s\sigma_g$ at low KER) are shown in blue ($2s\sigma_g$) and red ($2p\sigma_u$), others are not visible on that scale.”

To the caption of figure 3.

Reviewer:

On the other hand one may wonder if double ionization, which leads to a fast electron and a slow electron as such photon energies, could also partially contribute to the (H⁺,e) coincident events at the highest KERs considered here (14-22 eV): is such a contribution excluded here?

Our Response:

Yes, we exclude them from the dataset by making use of a photo ion photo ion coincidence map on the one hand and by exploiting momentum conservation on the other hand. For those double ionization events where we fail to detect the second proton due to detector efficiency we estimated and subtracted the resulting background. See figure A in reply to reviewer #3 above.

Reviewer:

The discussion of the results in Figure 2 clearly supports the developments given in the Supplementary Material, which provide the expression of the partial MFPADs associated with the assigned ionization channels in terms of the ground state HF orbitals representation in momentum space. They illustrate the statement (end of the first column of page 4) "It is solely due to electron correlation that the momentum distribution of electron A depends strongly on the properties of electron B".

This statement can also be thought from the photoionization perspective. If indeed it is solely due to electron correlation that channels involving ionization and excitation are populated, the $g \rightarrow u$ dipole selection rule plays a role in imposing an interdependence between the characteristics of electron A and electron B. The continuum molecular state being of u symmetry, the electronic wave function of electron A should be of g or u symmetry under the condition that electron B is detected in the $2p\sigma_u$ or $2s\sigma_g$ state of H_2^+ , respectively? This has consequences in terms of the different partial waves (with defined node properties) which contribute to the momentum distribution of electron A and it holds for the both the parallel and perpendicular transitions which add up in the observed MFPAD.

Could the authors comment briefly on this perspective, which is not so clear from eq (16-18)?

Our Response:

The reviewer is absolutely right in saying that selection rules play a role in dictating the form of the measured MFPADs. However, this comes as a consequence of electron correlation. Indeed, in the absence of electron correlation (HF level), the matrix elements given in the left hand side of eqs. (17) and (18) would be strictly zero. Only equation (16) survives. This is irrespective of any selection rule and therefore it is true for any operator O . Once electron correlation is plugged in, the actual contribution of eqs. (17) and (18) to the MFPAD is indeed the result of the dipole selection rules as explained by the reviewer. We have realized that in the paragraph mentioned by the reviewer, the use of the word "solely" is rather confusing. Therefore we have slightly modified the beginning of this paragraph, which now reads:

"Our experimentally obtained spectra not only show the imprint of correlation, but also allows us to separate the contribution of different pieces of the electronic wave function to this correlation. Indeed, the momentum distribution of electron A depends strongly on the properties of electron B."

We have also included a sentence in the Supplementary Material, after the discussion that follows eqs. (16) - (18) to emphasize that dipole selection rules are ultimately responsible for the actual shape of the MFPADs. It reads:

“It is worth noticing that the specific form of the MFPADs resulting from using eqs. (16)

-

(18) (or their simplified versions (21) - (23)) is the consequence of the dipole selection rule that operates in this particular problem. As a consequence, for transition operators \hat{O} different from the dipole one, different expressions would be obtained. Nevertheless, even in this case, one can anticipate that in the absence of electron correlation, the matrix elements given by eqs. (17) and (18) (or equivalently (22) and (23)) would be strictly zero.”

Reviewer:

In the discussion of Figure 4 describing the dependence of the momentum distributions and MFPADs with the internuclear distance (and KER through the reflection approximation), in addition to the similar trend found between the A to C momentum distributions and the D to F MFPADs, it is remarkable that looking at the computed MFPADs the highest resemblance is between A and E, and B and F. Any comment to be added in the text?

Our Response:

The shift between the plain wave result (A - C) and the full result (D - F) arises from the fact that we do not have fully reached the Born limit. Please see also G. L. Yudin, S. Chelkowski and A. D. Bandrauk, J. Phys. B: At. Mol. Opt. Phys. **39**, L17-L24 (2006).

Reviewer:

Methods

In the part devoted to the experiment it would be interesting that the authors provide more detail on the conditions enabling collection and momentum determination of the 360-380 eV electrons, at the core of this work: little is said about the electron arm of the spectrometer used and the cited references do not address explicitly such high energy conditions. The related resolution in momentum determination, and therefore in energy and angle, is of importance for the discussion of the measured MFPADs.

Our Response:

The methods section gives the technical details of our electron spectrometer: The electron arm of the spectrometer was formed by a 37 mm long acceleration region. The electrical field was 92 V/cm. Together with a magnetic field of 35.5 G parallel to the electric field, it guided the electrons on cyclotron trajectories towards the detector, consisting of a 80 mm time and position sensitive micro-channel plate with hexagonal delayline readout (see ref. [18]).

We measure an energy resolution of about 30 eV at 400 eV photon energy. This corresponds to a momentum resolution of approx. 0.2 a.u. From this we obtain an upper limit for our angular resolution of 2.3 degrees for the electron in the laboratory frame. The angular resolution in the molecular frame is further influenced by the precision of the alignment between the electron and the ion arm of the spectrometer. We estimate that the overall angular resolution in the molecular frame is better than 10deg.

Reviewer:

For a proper use of the count rate numbers given in this section (200 Hz of ions without the gas jet and 350 Hz with the gas jet), one should know which ions do the authors refer to? Are these numbers given for the H⁺ signal or for all ions?

Our Response:

This partially coincides with the request by reviewer #1. We have modified the according paragraph to clarify that these numbers are the total ion count rate over all charges states (see reply to reviewer #1).

REVIEWERS' COMMENTS:

Reviewer #1 (Remarks to the Author):

I have carefully read the manuscript and the response letter, and I am entirely satisfied with things in their present form. Based on that, I warmly do recommend this outstanding work for publication in Nature Communications.

Reviewer #2 (Remarks to the Author):

This is my second report of this paper. While other questions are replied precisely, I still didn't understand the circle in Fig 1(c). Sorry for the ambiguous description, the circle I mentioned is not the dashed line indicating the photoelectron momentum, but the very tiny circle with a radius about 1. Such a circle is also observed in Fig. 2 E, but not in Fig. 2 B, C, D, F.

Reviewer #3 (Remarks to the Author):

I found the changes to the manuscript to be thorough and appropriate. I am in favor of changing the title to the suggested new title.

Reviewer #4 (Remarks to the Author):

The points raised in the review process of the manuscript by M. Waitz et al have been satisfactorily addressed by the authors in their response letter and included in the revised manuscript, therefore I consider that the manuscript can be accepted for publication.

Below, we list our responses to the points raised by the reviewers:

- **Points raised by reviewer #1:**

Reviewer:

I have carefully read the manuscript and the response letter, and I am entirely satisfied with things in their present form. Based on that, I warmly do recommend this outstanding work for publication in Nature Communications.

- **Points raised by reviewer #2:**

Reviewer:

This is my second report of this paper. While other questions are replied precisely, I still didn't understand the circle in Fig 1(c). Sorry for the ambiguous description, the circle I mentioned is not the dashed line indicating the photoelectron momentum, but the very tiny circle with a radius about 1. Such a circle is also observed in Fig. 2 E, but not in Fig. 2 B, C, D, F.

Our Response:

The tiny circle with radius about one in figure 1 C and 2 E is a feature of the momentum space wave function that is plotted in these two subfigures. It's not artificially inserted to the plots, but an inherent pattern of the $2s\sigma_g$ wave function that gets visible when plotted in log scale. It is also present in the linear scale version of that plot in figure 1 B, but it's barely visible there.

- **Points raised by reviewer #3:**

Reviewer:

I found the changes to the manuscript to be thorough and appropriate. I am in favor of changing the title to the suggested new title.

Our Response:

The title has been changed to "Imaging the square of the correlated two-electron wave function of a hydrogen molecule"

- **Points raised by reviewer #4:**

Reviewer:

The points raised in the review process of the manuscript by M. Waitz et al have been satisfactorily addressed by the authors in their response letter and included in the revised manuscript, therefore I consider that the manuscript can be accepted for publication.